# I Know You Did Not Write That! A Sampling Based Watermarking Method for Identifying Machine Generated Text

## Abstract

Potential harms of Large Language Models such as mass misinformation and plagiarism can be partially mitigated if there exists a reliable way to detect machine generated text. In this paper, we propose a new watermarking method to detect machine-generated texts. Our method embeds a unique pattern within the generated text, ensuring that while the content remains coherent and natural to human readers, it carries distinct markers that can be identified algorithmically. Specifically, we intervene with the token sampling process in a way which enables us to trace back our token choices during the detection phase. We show how watermarking affects textual quality and compare our proposed method with a state-of-the-art watermarking method in terms of robustness and detectability. Through extensive experiments, we demonstrate the effectiveness of our watermarking scheme in distinguishing between watermarked and non-watermarked text, achieving high detection rates while maintaining textual quality.

## 1  Introduction

Transformer based Large Language Models (LLMs) (Vaswani et al., 2023) such as ChatGPT, Llama2 (Touvron et al., 2023) are able to generate texts that closely resemble human authored texts. For instance, Clark et al. (2021) report that untrained humans are not able to distinguish between texts generated by GPT-3 and texts authored by humans. As we train larger models with more parameters on an ever-expanding corpora, their capabilities in generating human-like text are likely to increase (Hoffmann et al., 2022). With their incredible performance in text generation, they become effective tools for automating monotonous text based tasks such as summarization and translation (Radford et al., 2019).

However, these LLMs pose various threats to society because they can be also used for bad causes such as generating credible-sounding misinformation (Pan et al., 2023), creating fake product reviews (Adelani et al., 2019) and academic plagiarism (Dehouche, 2021). Recent studies have discovered that even though LLM-generated responses may sound convincing, they can be frequently incorrect (Lin et al., 2022).

The potential negative consequences associated with LLMs can be reduced significantly if a reliable detection system is in place to differentiate between machine-generated and human-written texts. A number of researchers focused on this important problem and proposed various approaches training a classifier (Guo et al., 2023), detecting based on linguistic features (Guo et al., 2023) and log probabilities and perturbations (Mitchell et al., 2023). Data driven methods such as training classifiers requires wide range of data with different styles, sources, and languages. More importantly non-watermarking detectors are generally found to be ineffective (Krishna et al., 2023). Moreover, existing perplexity based detectors are biased against non-native English writers (Liang et al., 2023), raising ethical concerns about their usage in real-life applications.

In this paper we propose a novel model-agnostic watermarking method to detect machine generated text. In watermarking, a hidden pattern is inserted to a passage that is imperceptible to humans but can be easily detected an algorithm.

In our proposal, we interfere with the randomness of sampling a new token to be generated in the decoding phase of LLMs. For each token to be generated, we sample multiple candidate tokens based on their probability provided by the LLM and calculate a *secret number* for each of the candidate tokens. Subsequently, we pick the token with the highest secret number value. The way we calculate the secret number enables us to retrieve the same values from generated text. And our maximization effort lets us discriminate against non-watermarked text.

In our experiments, we evaluate the quality of the watermarked texts and how accurately we can detect the watermarks using various datasets and LLMs. We also compare our model against watermarking method of Kirchenbauer et al. (2023a). In our experiments, we show that we are able to detect watermarked texts almost in all cases. In addition, we observe that our method based on sampling with replacement does not reduce the text quality in almost all cases while our method based on sampling without replacement yields slight decrease in text quality. In addition, we show that our proposed method is robust to token level paraphrasing attacks.

The main contributions of our work are as follows. i) We introduce a novel watermarking scheme to detect machine-generated text. In our comprehensive evaluation we show that our watermarks are highly detectable while causing a slight decrease in text quality. ii) We share both our code and dataset to ensure reproducibility of our results and help other researchers build upon our findings.[1].

## 2  RELATED WORK

The remarkable achievements of Large Language Models (LLMs) compelled researchers to shift their attention towards understanding their potential drawbacks and risks. We direct readers to the survey studies conducted by Crothers et al. (2022) and Weidinger et al. (2021) for an in-depth analysis of the risks associated with LLMs. Now, we focus on studies on detecting texts generated by LLMs.

### 2.1  NON-WATERMARKING DETECTION METHODS

Gehrmann et al. (2019) propose a tool GLTR which works in a white-box setting and highlights texts based on probability distribution of tokens provided by the LLMs. They show that their visual tool improves the human detection rate of machine generated text from 54% to 72% without any prior training and without tampering with the text generation phase.

Mitchell et al. (2023) also work in a white-box setting and create perturbations of the candidate text and analyze the negative curvature regions of the model's log probability function. Their main hypothesis for detection is as follows. When machine generated text is modified it tends to have lower log probability. However, modifications on the human-written text may have higher or lower log probability than the unmodified text.

Zellers et al. (2019) examine several schemes to detect fake news article using GROVER which is a language model that generates and classifies fake news articles. They conclude that the most effective model for identifying fake news generated by GROVER is the model itself. Adelani et al. (2019) also report that GROVER is highly accurate in detecting fake reviews. Zellers et al. (2019) argue that machine-generated text classification requires a similar inductive bias as the generator model, rather than expressive capability. However, these findings differ from those of Solaiman et al. (2019) as they claim that a fine-tuned RoBERTa model is a more effective detector than a similarly-capable fine-tuned GPT-2 model.

A number of researchers focused on developing machine learning models to identify generated texts. For instance, Fagni et al. (2021) report that transformer based classifiers to be the best discriminators of fake tweets. Guo et al. (2023) collect a dataset of ChatGPT's and experts' responses to questions in various topics such as finance and medicine. and train classifiers to predict if a given passage is computer generated. A similar approach is also followed by the creators of ChatGPT with underwhelming results[2]. In our work, we propose a watermarking method to detect generated texts. We discuss watermarking methods in the literature in the following section.

---

[1]The URL is hidden due to the double blind review process

[2]https://openai.com/blog/new-ai-classifier-for-indicating-ai-written-text/

## 2.2 WATERMARKING DETECTION METHODS

Abdelnabi and Fritz (2020) introduce Adversarial Watermarking Transformer (AWT) model, which encodes binary messages in text to trace its origin and prevent malicious use, using a jointly trained encoder-decoder and adversarial training, ensuring the watermark is discreet while maintaining the text's original meaning. Ueoka et al. (2021) proposes using a masked language model, which has a high payload capacity and is less susceptible to automatic detection than generation-based methods.

The closest work to our own is Kirchenbauer et al. (2023a)'s watermarking method. They propose selecting a randomized subset of approved tokens from the vocabulary and then promoting the sampling of the tokens from chosen approved subset of the vocabulary via increasing the subsets logits. The randomization is seeded on previously generated token(s) in a context window. In our work, we interfere the sampling process without changing LLMs' probability distribution over vocabulary while Kirchenbauer et al. (2023a) interfere the probability distribution. In our experiments, we extensively compare our proposed method against Kirchenbauer et al. (2023a)'s.

Watermarking methods are investigated to protect text generation models against model extraction attacks, Zhao et al. (2023) inject a sinusoidal signal into the model's generation probabilities. As a result, the suspect model is anticipated to retain the watermark from the victim model, rendering it detectable.

## 2.3 PARAPHRASING ATTACKS

As there are tools to detect generated texts, people might want to avoid these detection tools by intentionally changing the generated texts. Therefore, prior work also explored how vulnerable detection systems are against paraphrasing attacks.

Sadasivan et al. (2023) demonstrate how effective off-the-shelf sentence-level paraphrasing models can be at evading detection and conclude that detecting generated text is an unsolvable problem. However, this conclusion is contradicted by Chakraborty et al. (2023) as they show that detection should always be possible when there exist enough samples. Krishna et al. (2023) develop a paraphrasing model which successfully evades several detectors including watermarking (Kirchenbauer et al., 2023a) and DetectGPT (Mitchell et al., 2023). They conclude that non-watermarking detectors to be generally ineffective. In their proposed detection scheme, the API provider maintains a database containing every sequence generated by their LLM. When a detection query is initiated, this database is queried to identify a previously-generated sequence that exhibits the highest semantic similarity to the query. If the level of similarity surpasses a predefined threshold, the query is classified as machine-generated.

## 3 PROBLEM DEFINITION

Our goal is to develop a model-agnostic watermarking method to identify generated texts. Let $LLM$ be a large language model and $LLM^w$ is its version with watermarking feature. In addition, let $T_{LLM}(P)/T_{LLM}^w(P)$ be a text generated by $LLM/LLM^w$ for the given prompt $P$. An ideal watermarking method should have the following properties:

- The watermarking process should not decrease the quality of the texts, i.e., the quality of $T_{LLM}(P)$ and $T_{LLM}^w(P)$ should be similar for any given $P$.
- Watermarking text should not necessitate retraining or fine-tuning.
- We should have the capability to compute a statistical confidence interval with interpretable values for the detection and sensitivity analysis of the watermark.
- The watermark should be robust to perturbations. An adversary must make significant modifications to remove the watermark.

## 4 PROPOSED METHODOLOGY

In this section, we explain our proposed method to generate watermarked text (Section 4.1) and how to detect the watermark within a given text (Section 4.2).

## 4.1 Generating Watermarked Texts

In our watermarking method, we interfere with the randomness of picking the next token according to its conditional probability provided by a language model in the decoding stage. The details of our method are shown in **Algorithm 1**.

---

**Algorithm 1** Text Generation with the Sampling Watermarker

---

    **Input:** P {Prompt given to the model}
    **Parameter-1:** $y$ {The sampling count}
    **Parameter-2:** $k$ {The context window size}
1: $T_{LLM}(P) = P$ {Keeps the whole text}
2: **for** each token to be generated **do**
3:     $D = LLM(T_{LLM}(P))$ {Get the probability distribution from the LLM}
4:     $C_{[1-y]} = sample(D, y)$ {Sample y candidate tokens}
5:     **for** $i \in \{1, \ldots, y\}$ **do**
6:         $S^{C_i} = RNG(seed = hash(T_{LLM}(P)^{[N,N-k]}, C_i))$ {Calculate the secret number}
7:     **end for**
8:     $T_{LLM}(P) = T_{LLM}(P) + C^{argmax(S^{C_1}, \cdots, S^{C_y})}$ {Concatenate the selected token}
9: **end for**

---

For a given input prompt $P$, $LLM$ produces a text T in an iterative way [Lines 1-9]. In each iteration, LLM outputs a conditional probability distribution vector over the vocabulary $V$ for the next token to be generated [Line 3]. We multinomially sample $y$ candidate tokens based on the probability distribution vector [Line 4]. Subsequently, we compute a *secret number* for each candidate token $t$ [Lines 5-7]. In order to compute the secret number of a candidate token ($S^t$), we first concatenate the $k$ previous tokens and the candidate token $t$ and then calculate their SHA256 hash value. Subsequently, we seed a random number generator with the hash value [Line 6] and generate a random number. Next we pick the token with the highest secret number for the next token [Line 8].

The secret number of any token in a candidate passage only depends on itself and the $k$ tokens that precede it. This enables us to retrieve the same secret number for every token in a passage outside of the generation process. Moreover, if a passage is watermarked we expect the average secret number of the tokens that make up the text to be significantly higher than otherwise. This is because while the production of the non-watermarked text is completely ignorant of the secret numbers of tokens, our watermarking scheme actively attempts to maximize this value.

During sampling, we have the option to sample candidate tokens with or without replacement. When we sample without replacement, the secret numbers of the candidate tokens are guaranteed to be distinct values. Maximizing the use of distinct values tends to result in larger secret number values, making the watermark more detectable. On the other hand, if the entropy of the probability distribution is low, i.e., there are few plausible tokens to be generated, sampling without replacement would cause the model to pick the unlikely tokens, reducing the quality of the generated text. Therefore, we also explore sampling with replacement and evaluate the impact of both sampling methods in Section 5.

## 4.2 Detecting the watermark

In order to detect whether a given text X is watermarked or not, i.e., a text generated by our scheme or not, we first tokenize X and calculate the secret number of each token in X. The secret number of the $r^{th}$ token of X can be calculated as follows.

$$S^{X_r} = RNG(seed = hash(X_{(r-k)} \cdots X_{(r-1)}, X_{(r)}))$$
(1)

where $RNG$ is a random number generator which draws values from a continuous uniform distribution spanning the interval from zero to one. The anticipated mean of the secret number for the tokens composing a text aligns with a normal distribution characterized by an expected mean of 0.5 and an expected variance of $\frac{1}{12*N}$ (See Blitzstein and Hwang (2015) for explanation), where $N$ represents the number of tokens within the given text X. As the length of the candidate text increases, the average secret number for non-watermarked text gradually approaches this theoretical distribution

with diminishing variance, thus reducing the likelihood of the text's average secret number deviating significantly from 0.5. Conversely, during the watermarking process, tokens are selected from a set of candidates based on their possession of the highest secret number (out of $y$ candidates). This selection dramatically alters the distribution of the average secret number, rendering it exceedingly improbable for the text to have arisen through natural generation. Thus, we classify the text as watermarked if a certain threshold is exceeded. Formally, we define the following null hypothesis.

$H_0$: *The text sequence is generated without any attempt to maximize the secret number average.*

The formula of the z-score for testing the hypothesis is as follows:

$$z - score = (\overline{sna} - 0.5)/\sqrt{1/(12 \cdot N)} \tag{2}$$

where $\overline{sna}$ denotes the secret number average of the candidate text and $N$ represents how many tokens make up the candidate text. The null hypothesis is rejected (and the watermark is detected) if $z - score$ is above a chosen threshold $u$.

## 5 EXPERIMENTS

### 5.1 EXPERIMENTAL SETUP

In this section, we explain evaluation metrics (Section 5.1.1) to assess the quality of our watermarking method, describe the models we used for watermarking (Section 5.1.2), baseline methods we compare against our methods (Section 5.1.3), and datasets we utilized in our experiment (Section 5.1.4). Lastly, we provide details about implementation details (Section 5.1.5).

### 5.1.1 EVALUATION METRICS

In order to measure the quality of watermarking methods, we focus on the quality of the generated text and our detection rate. We adopt the measures used by related prior work (Kirchenbauer et al., 2023b; Krishna et al., 2023). In particular, we calculate how the generated texts are similar to the human authored ones using *P-SP* (Wieting et al., 2023). In addition, we use *diversity* which aggregates n-gram repetition rates. A high diversity score represents a more diverse text where fewer n-grams are repeated (Li et al., 2023). Given the fraction of unique $n$-grams (which is denoted as $u_n$) diversity up to the $N^{th}$ order is defined as follows.

$$\text{diversity} = -\log \left( 1 - \prod_{n=1}^{N} (1 - u_n) \right) \tag{3}$$

Lastly, we use *coherence* to measure the semantic coherence between the prompt and the generated text. We employ the sentence embedding method, SimCSE (Gao et al., 2022) for this calculation. Given the prompt $x$ and the generated text $\hat{x}$, the coherence score is defined as $v_x^\top v_{\hat{x}}/(\|v_x\| \cdot \|v_{\hat{x}}\|)$, where $v_x = \text{SimCSE}(x)$ and $v_{\hat{x}} = \text{SimCSE}(\hat{x})$.

### 5.1.2 MODELS

As our approach can be applied in any model, we utilize three different models that our hardware systems could execute. In particular, we use OPT (Zhang et al., 2022) with 1.3B parameters, BTLM-3B (Dey et al., 2023) with 3B parameters, and Llama2 (Touvron et al., 2023) with 7B parameters.

### 5.1.3 BASELINE METHODS

We compare our proposed method against Kirchenbauer et al. (2023a)'s study, which is also known as "Maryland Watermark" (MWM). For the configuration parameters of their approach, we set the greenlist fraction $\gamma$ at 0.25 and set the logit bias $\delta$ to 2.

### 5.1.4 DATASETS

In our experiment, we use two different datasets: i) the train split of the 'realnewslike' portion of the C4 (stands for "Colossal Clean Crawled Corpus") dataset (Raffel et al., 2020) and ii) the train

split for Wikitext (103-v1-raw) dataset (Merity et al., 2016). C4 is an extensive web text collection resembling real news articles while Wikitext consists of 100M tokens extracted from the set of verified *Good* and *Featured* articles on Wikipedia, providing a more structured and manageable source.

We use the first 100 tokens of the passages as prompts. In order to have a fair comparison, we use 200 tokens for all cases. Therefore, we allow models to generate maximum 200 new tokens. For a given prompt, if any of the generated text is less than 200 tokens, we discard it, and try another prompt drawn from the corresponding dataset. We continue this process until we reach 500 samples for each dataset. Eventually, for each dataset and model we use, we create five text subdatasets: i) texts generated by Maryland watermarking ($T_{MWM}$), ii) texts generated by our approach with sampling with replacement ($T_{SWR}$), iii) texts generated by our approach with sampling without replacement ($T_{SWOR}$), iv) texts generated without watermark ($T_{NoWM}$), and v) texts authored by humans ($T_{Humans}$).

### 5.1.5 IMPLEMENTATION

We implemented the sampling watermarker using the PyTorch (Paszke et al., 2019) backend of the Hugging Face library (Wolf et al., 2019). We utilized the `generate` API provided by Hugging Face for generating text. This API allows for passing a custom `LogitsProcessor` which can be used to modify the prediction scores of a language model head for generation. We use Top-k sampling (Fan et al., 2018) with $top-k = 40$ before doing any sampling on all methods. For our proposed method we set the context window size $k$ to 1 and sampling count $y$ to 5 unless otherwise is mentioned.

## 5.2 EXPERIMENTAL RESULTS

This section comprises of four subsections, each serving distinct research objectives. The first (Section 5.2.1) assesses watermark detectability, the second (Section 5.2.2) examines textual quality under watermarking, the third (Section 5.2.3) evaluates watermark robustness against attacks, and the final subsection (Section 5.2.4) investigates the impact of various generation parameters on watermarking performance.

### 5.2.1 DETECTIBILITY EXPERIMENTS

In this experiment, we assess how accurate watermark detection mechanisms work. Specifically, we run our watermarking methods and MWM for all datasets we create and calculate average z-scores over the generations. In addition, we set the z-score threshold ($u$) to 4 for both watermarking schemes as in (Kirchenbauer et al., 2023a) and calculate the percentage of the texts detected as watermarked. The results are shown in **Table 1**.

| | | C4 | | | | | | Wikitext | | | | | |
| | | OPT-1.3B | | BTLM-3B | | Llama2-7B | | OPT-1.3B | | BTLM-3B | | Llama2-7B | |
| Text | Detector | z-score | %WM | z-score | %WM | z-score | %WM | z-score | %WM | z-score | %WM | z-score | %WM |
|---|---|---|---|---|---|---|---|---|---|---|---|---|---|
| $T_{SWR}$ | SWR | 11.31 | 99.8% | 10.11 | 99.8% | 9.44 | 99% | 12.09 | 99.8% | 10.33 | 100% | 10.36 | 99.8% |
| $T_{SWOR}$ | SWOR | **16.85** | 100% | **16.29** | 100% | **16.66** | 100% | **16.92** | 100% | **16.26** | 100% | **17.23** | 100% |
| $T_{MWM}$ | MWM | 10.77 | 100% | 9.82 | 100% | 9.71 | 99.4% | 11.79 | 100% | 10.43 | 100% | 10.65 | 97% |
| $T_{Humans}$ | SWR | 0.27 | 0% | -0.07 | 0% | 0.22 | 0% | 0.03 | 0% | -0.05 | 0% | 0.28 | 0% |
| | MWM | -0.23 | 0% | -0.46 | 0.2% | 0.21 | 0.2% | 0.35 | 0.6% | 0.21 | 0.2% | -0.01 | 0.2% |
| $T_{NoWM}$. | SWR | 0.22 | 0% | -0.25 | 0% | 0.44 | 1.4% | 0.69 | 0.6% | -0.22 | 0% | 0.17 | 3.6% |
| | MWM | -0.25 | 0% | -0.42 | 0.2% | 0.32 | 1% | 0.01 | 0.4% | -0.17 | 0.2% | 0.39 | 3.4% |

Table 1: The average z-scores over the generations when attempted to detect the watermark and the ratio of samples detected as "watermarked" by the corresponding detector. The text in **bold** represent the highest z-score for watermarked text and lowest for baseline completion text.

The average z-scores exceed 10 in most of the watermarked texts, and is near 0 for non-watermarked text, showing the effectiveness of watermarking schemes. SWOR achieves achieves the highest z-score and detection rates in watermarked texts.

Our watermarking methods consistently avoid false positives when applied to human authored text, whereas MWM occasionally misidentifies such content as watermarked. Moreover, both MWM and our approach have higher false positive when dealing with non-watermarked machine-generated

text compared to human authored text. This is because non-watermarked machine-generated text inherently resembles watermarked machine-generated text.

### 5.2.2 TEXTUAL QUALITY EXPERIMENTS

In this experiment, we assess how watermarking affects the textual quality. We report P-SP, diversity, and coherence scores in in **Table 2** for texts watermarked with our approaches, Maryland Watermarking, and without any watermark. Additionally, for a more comprehensive understanding, we provide samples of watermarked texts in the **Appendix A**.

Regarding similarity with respect to human authored text (P-SP), we observe that MWM achieves higher scores than our methods for OPT-1.3B and BTLM-3B. However, SWR outperforms others when Llama2-7B is used for generation. Interestingly, SWR even yields higher P-SP score than non-watermarked text with Llama2-7B in Wikitext. We observe a similar pattern in other metrics such that MWM yields higher score with OPT-1.3B and BTLM-3B models than our models in most of the cases. On the other hand, SWR outperforms others with the largest model we use. Regarding SWOR vs. MWM with Llama2-7B is mix such that SWOR outperform MWM in Wikitext but not in C4.

| Metric | Method | C4 | | | Wikitext | | |
|---|---|---|---|---|---|---|---|
| | | OPT-1.3B | BTLM-3B | Llama2-7B | OPT-1.3B | BTLM-3B | Llama2-7B |
| P-SP | SWR | 0.44 | 0.48 | **0.48** | 0.45 | 0.48 | **0.52** |
| | SWOR | 0.40 | 0.42 | 0.38 | 0.42 | 0.43 | 0.44 |
| | MWM | **0.46** | **0.49** | 0.45 | **0.47** | **0.49** | 0.41 |
| | NWM | 0.47 | 0.50 | 0.48 | 0.49 | 0.49 | 0.46 |
| Diversity | SWR | 6.92 | 7.50 | **8.16** | 6.26 | 7.06 | **6.96** |
| | SWOR | 6.84 | 7.49 | 7.48 | 6.42 | 7.23 | 6.66 |
| | MWM | **7.40** | **7.90** | 5.88 | **6.77** | **7.46** | 5.38 |
| | NWM | 7.87 | 7.87 | 6.17 | 7.16 | 7.55 | 6.1 |
| Coherence | SWR | 0.63 | **0.64** | 0.64 | 0.67 | **0.66** | **0.65** |
| | SWOR | 0.58 | 0.59 | 0.53 | 0.63 | 0.60 | 0.54 |
| | MWM | **0.64** | **0.64** | **0.65** | **0.68** | **0.66** | 0.58 |
| | NWM | 0.66 | 0.66 | 0.67 | 0.70 | 0.66 | 0.62 |

Table 2: The impact of watermarking on the the quality of the generated text. The highest score among watermarked texts for each case is shown in **bold**. MWM: Maryland Watermarking, SWR: Sampling with replacement, SWOR: Sampling without replacement, NWM: No Watermarking.

### 5.2.3 ROBUSTNESS EXPERIMENTS

In order to assess how vulnerable the watermarking methods are against token level paraphrasing attacks, we conduct an experiment similar to the one in (Kirchenbauer et al., 2023a). In particular, we randomly pick $\%t$ of tokens in the watermarked and mask them. Next, we use DistilRoBERTa-Base model (Sanh et al., 2020) to replace masked tokens, ensuring that the model did not predict the same token that was initially masked. **Figure 1** shows how different attack percentages effect the detection of the watermarked text. Sampling without replacement achieves high detection rates even in attacks with %40, outperforming all other methods. Sampling with replacement and Maryland Watermarker achieve similar detection rates.

### 5.2.4 THE IMPACT OF SAMPLING COUNT

We explore the impact of the sampling count used for secret number generation, $y$ on the quality of the generated texts and the detection rate. In particular, we vary $y$ from 2 to 11 and generate text using our approach with and without replacement using C4 dataset and Llama-2-7B model. **Table 3** shows the text quality metrics along with average z-score and detection rate. We observe that increasing the sampling count $y$ results in decreasing quality scores in all cases, but yields higher z-scores. Detection rate for SWOR remains at %100 even at a low sampling count of $y = 2$ and SWR achieves 99% rate when $y = 5$.

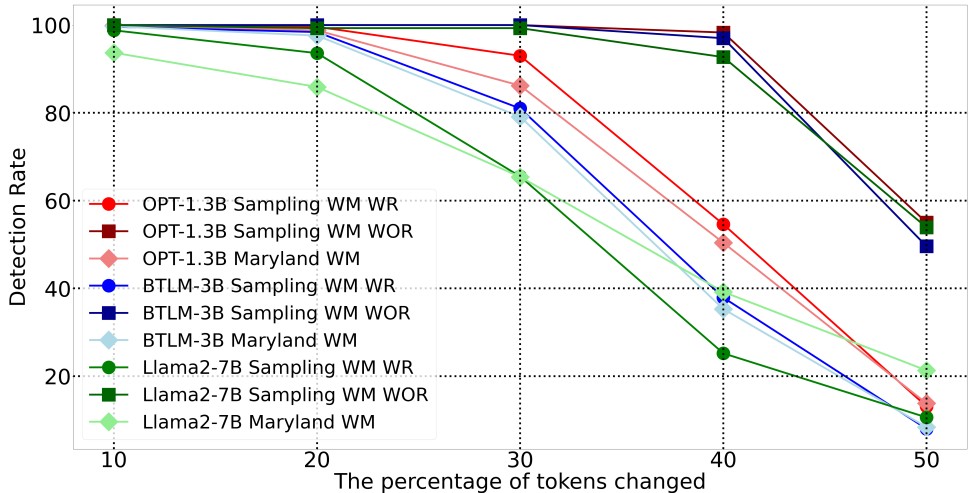

Figure 1: The impact of paraphrasing attacks on the detection rate of watermarked texts.

| $y$ | P-SP | | Diversity | | Coherence | | z-score | | Detection Rate | |
|---|---|---|---|---|---|---|---|---|---|---|
| | SWR | SWOR | SWR | SWOR | SWR | SWOR | SWR | SWOR | SWR | SWOR |
| 2 | 0.49 | 0.45 | 8.33 | 8.65 | 0.66 | 0.61 | 4.79 | 8.33 | %76 | %100 |
| 5 | 0.48 | 0.38 | 8.16 | 7.48 | 0.64 | 0.53 | 9.44 | 16.66 | %99 | %100 |
| 8 | 0.46 | 0.34 | 7.66 | 6.4 | 0.62 | 0.50 | 11.72 | 19.51 | %100 | %100 |
| 11 | 0.45 | 0.30 | 7.65 | 5.83 | 0.62 | 0.46 | 12.91 | 20.94 | %100 | %100 |

Table 3: The effect of sampling count $y$ on textual quality metrics. Model: Llama-2-7B, Dataset:c4, $k$:1.

### 5.2.5 ENTROPY IN PROBABILITY DISTRIBUTION

The effectiveness of our proposed method and the Maryland watermarking depends on the language model' output distribution. For instance, if the model outputs a low entropy distribution for the next token, our sampling with replacement based method is likely to sample the same $y$ tokens as candidates. However, in sampling without replacement case, the watermarker is guaranteed to sample $y$ unique tokens and pick the one that has the highest secret number.

In this experiment, we manually manipulate the output distribution entropy of our models by adjusting the sampling temperatures to assess its impact. **Table 4** shows the average z-score for varying temperature values for Llama2-7B model on C4 dataset. As expected we observe that both SWR and MWM exhibit stronger watermarks when the output distribution entropy is higher. SWOR shows slight variations in the average z-score but these are just statistical noises as SWOR is designed to be unaffected by the underlying distribution entropy.

| Temperature | 0.8 | 0.9 | 1 | 1.1 | 1.2 |
|---|---|---|---|---|---|
| **SWR** | 8.14 | 8.91 | 9.44 | 10.38 | 10.82 |
| **SWOR** | 16.89 | 16.75 | 16.66 | 16.68 | 16.61 |
| **MWM** | 8.02 | 8.85 | 9.71 | 10.65 | 11.24 |

Table 4: The effect of sampling temperature on the average z-score. Lower temperatures yield output distributions with lower entropy vice versa. Model: Llama2-7B, Dataset:C4, $k$:1,$y$:5

## 6 LIMITATIONS

While our work makes an important contribution in the research of LLMs, there there are certain limitations that require further research in the future. Firstly, we derive our prompts from two

different datasets. However, the watermarking performance highly depends on the given prompt. For instance, if we ask a factual question to a model (e.g, what is the full text of the U.S. constitution?), watermarking the generated output would be challenging because of limited flexibility in the answer. Therefore, a larger number of datasets covering diverse set of topics are required. Similarly, we ran our experiments for three models due to hardware limitations. As the watermarking performance is affected by the models used for text generation, covering a wider range of LLMs is required for a more reliable assessment of the methods.

Furthermore, in our study, we focus on only the task of completing a text for a given prompt. We acknowledge that further evaluation of the proposed watermark across different down stream tasks such as question answering and summarization would be beneficial. We leave this exploration as future work.

Lastly, we explore only token level paraphrasing attacks to measure the robustness of the models. There exist different methods for manipulating text to evade watermarking detection such as deletion, unicode attacks and human paraphrasing. Thus, other types of attacks should be explored to further analyze the robustness of watermarking methods.

## 7 CONCLUSION AND FUTURE WORK

In this work, we propose a watermarking scheme which embeds a unique pattern into the generated text while preserving its coherence and natural readability for human readers. Specifically, We modify the token sampling process of LLMs. In particular, we first sample multiple tokens based on probability distribution over vocabulary and then calculate a unique secret number for each sampled one. We always pick the token with the highest secret number, allowing us to trace the hints of generation process.

In our experiments with multiple datasets and LLMs, we show that our method we show that our watermarking is detectable and reduce slight decrease in text quality. Furthermore, our method outperforms Kirchenbauer et al. (2023a)'s method in terms of detectability and robustness. Regarding text quality, we achieve slightly superior results compared to (Kirchenbauer et al., 2023a) when applied to larger models, albeit with less favorable outcomes when dealing with smaller models.

There are multiple research directions we plan to extend in the future. Firstly, we plan to conduct our experiments on a larger scale in terms of data and model size and types. Secondly, a more sophisticated watermark could be implemented by adaptively choosing the sampling count $y$ based on the entropy of the output distribution. Specifically, when the output distribution exhibits low entropy, we can select a smaller value for $y$ and conversely, when the entropy is high, we can opt for a larger value. This method would ensure less perplexity on low entropy text while allowing for a stronger watermark to be embedded on higher entropy text. We leave this extension as a future work.

Lastly, there are no inherent obstacles to abstaining from the concurrent application of both our and Kirchenbauer et al. (2023a)'s watermarks during text generation. This would enable texts that are detectable by both watermarking methods. Employing two relatively less intrusive watermarks might potentially better maintain the textual quality while preserving high detectability.

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

## A    SAMPLE GENERATED TEXTS

| prompt | real completion | no watermark | Maryland watermarked | Sampling Watermarked(WR) | Sampling Watermarked(WOR) |
|---|---|---|---|---|---|
| ...Deputy Short radioed for backup, drew his firearm and commanded Zubrod to step back from the victim. Zubrod initially complied by stepping back and dropping the hammer. Deputy Short holstered his gun and took out his | TASER. Zubrod responded by saying something about finding a gun and began to reach down under the bed. He came up empty-handed, grabbed a nearby pair of scissors and immediately stabbed Schukei in the neck while she was lying on the floor.\n Deputy Short fired his TASER but it was not effective in incapacitating Zubrod[...continues] | handcuffs. As he turned to go back to the house, Zubrod came at him and punched him in the face.\n Deputy Short, who was not wearing a bulletproof vest, fired his firearm, striking Zubrod in the chest and killing him[...continues] | handcuffs. \n When Deputy Short was within reach, Zubrod suddenly swung the hammer and hit him on the side of the head, causing severe injury to the deputy. Deputy Short was hospitalized for five days, underwent a number of surgeries, including a craniotomy, and suffered extensive brain trauma.[...continues] | taser and pointed it at Zubrod. Deputy Short did not know that the hammer had been de-cocked at that point. Deputy Short later claimed that the hammer was de-cocked. Deputy Short claimed that his firearm discharged once in the bedroom area, once in the bathroom area[...continues] | patrol binder, direct traffic from a motorized ATV with no lights on or turn-on indicator. Deputy Layne Stuckey responds to assist deputize the situation and back Deputee' Short off from continuing the investigation. Deputee' Stubke asks where I'm from. I respond the "Same City". [...continues] |
| ...They didn't say all wildlife management is based on proven science, not guesswork and theories by people who oppose hunting. Thirty-three states have bear hunting, and Missouri will probably have bear hunting when the bear population reaches 500.\n New Jersey bear population is estimated at about 2, | 400, probably more than that from the cubs born last January. The most densely state–ours–and the second smallest in size.\n And the Show-Me State with 68,700 square miles worries about bears when the population reaches 500! New Jersey has 8,722 square miles with too many bears.\n New Jersey hunters never demanded bear hunting.[...continues] | 000, and the bear hunt is set for Oct. 12-14.\n The antihunters don't want the hunt, even though the bear population is growing, and they want to put the state's bear population on a diet.\n The state has about 2,000 acres of land set aside for the bear population, and the antihunters want to make it larger.[...continues] | 000. \n This year's bear hunting will be a lottery.\n Antihunters said it's cruel to kill bears.\n So it is cruel to kill animals that can't kill us, like mice, deer, geese, turkeys, etc. \n Antihunters said it's cruel to kill animals that can't kill us, like mice, deer, geese, turkeys, etc. \n Bears kill people.[...continues] | 500-3,000 bears.\n In Missouri, 11% of the bear population was taken last year.\n And there have been no cases this fall that I know of of people getting mauaued by bears.\n In their rush for an anti-hunting victory, they are hurting the bears.[...continues] | 512.\n Think that is too low a number for this size area for beer. A biased count was done by the DFWA (Don Foschi Alpha Male Animals), whose members like Fisk (Fox-Skate or Fox-Guess). Guesses based on a low population. And what do they propose, an airplane flyoever to count.[...continues] |
| ...It passed 68-30, with a group of 27 Democrats and one independent, mostly from states where gun rights are widely supported, joining all 40 Republicans in voting for the measure.\n Under the Wicker amendment, | Amtrak would lose the funds earmarked for it in the must-pass spending bill if it did not comply with the new regulations. In a statement released after the amendment passed, Mr. Wicker stressed that the guidelines laid out in the provision[...continues] | the federal government would be prohibited from withholding federal grants from states that do not allow people with concealed-carry permits to carry their guns across state lines.\n The amendment, which was attached to a $494 billion spending bill for [...continues] | the National Park Service, the Fish and Wildlife Service and the Bureau of Land Management would be banned from enforcing any federal regulation "relating to the regulation of the use, possession, transfer, receipt, shipment or surplus of any firearm, ammunition[...continues] | ...18 to 20 gun shows held year-round by fairgrounds or nonprofit shooting clubs in major cities would no longer have to require a special license.\n Those currently required are in places like Staten Island, N.Y.; Portland, Ore.; and Birmingham, Ala. And they are often required for only certain days of events.[...continues] | guns could also not law enforcers to keep guns with gun-restriction laws and rules impende.A version of this article first published September27. Read it on npr org. Photo Credite. Nicolette Barr. Nicolette spent a week on safait.[...continues] |
| ...Saturday's 2-2 draw with Oxford United saw Barton's 10th-placed team switch from 4-4-2 to 4-3-3, the formation he says he wants to play from now on as Town prepare to travel | to Rochdale this weekend.\n The head coach said: "The key for me is how do we progress and move forward. We changed system on Saturday.\n "This is how we will be playing going forward. I think it is the correct way to play.[...continues] | to Portsmouth on Saturday.\n Barton said: "It's here to stay. We'll look to play that formation for the rest of the season.\n "I think we've got the players to play that way.\n "We've got a good squad. We've got a good mix of youth and experience.[...continues] | to Oldham Athletic on Tuesday.\n Town were two goals up in the first half before dropping off and allowing Oxford to fight back and, when the visitors went in 2-2, it felt like it might have been a bad idea.\n Barton was disappointed that Town[...continues] | to Plymouth Argyle in League One on Saturday.\n Former central midfield partners Ephron Mason-Randolph and Lewis Evans were shifted to wide roles on the right hand-side and former captain Jimmy-Bell found his place, too, as he[...continues] | across to Blackwell Mewse on January.\n And in the first instance Town lined-up in five changes as Ben Lindsard restored and both Craig Mackrell - after ankledown injury[...continues] |

Table 1A: Some generation samples for watermarks. Model:Llama-2-7B, Dataset:c4, $y$:5, $k$:1, Temperature:1.0