# OpenReview forum: "I Know You Did Not Write That! A Sampling Based Watermarking Method for Identifying Machine Generated Text"
_ICLR.cc/2024/Conference — Submitted to ICLR 2024_

### Official Review · Reviewer_gCgZ · 2023-10-31

**Soundness:** 3 good
**Presentation:** 3 good
**Contribution:** 3 good
**Rating:** 5
**Confidence:** 4

**Summary:**

This paper addresses the challenge of identifying texts generated by Large Language Models (LLMs), such as ChatGPT, in the context of the potential harms posed by machine-generated misinformation and plagiarism. The authors propose a novel watermarking method aimed at embedding a unique, algorithmically identifiable pattern within machine-generated texts. Unlike existing methods, this approach intervenes in the token sampling process during text generation, ensuring that the generated content remains coherent and natural to human readers while carrying distinct, detectable markers.

The proposed watermarking method is model-agnostic and robust against token-level paraphrasing attacks. Through extensive experiments, the authors demonstrate the effectiveness of their approach, showing that it can accurately detect watermarked texts in almost all cases without significantly compromising the textual quality.

**Strengths:**

The major strengths of the paper include:

1. Robustness to Attacks: The proposed watermarking method has been designed to be robust against token-level paraphrasing attacks, ensuring that the watermark remains detectable even when parts of the text are altered.

2. Model-Agnostic Approach: The watermarking method is model-agnostic, meaning it can be applied across various Large Language Models (LLMs), making it versatile and widely applicable.

3. Comprehensive Evaluation: The authors conducted extensive experiments to evaluate the effectiveness of their watermarking scheme in distinguishing between watermarked and non-watermarked text, achieving high detection rates while maintaining textual quality.

**Weaknesses:**

1. Limited Exploration of Attacks: The paper primarily focuses on token-level paraphrasing attacks for evaluating the robustness of the watermarking method. Other types of attacks, such as deletion, unicode attacks, and human paraphrasing, are mentioned as areas for future exploration.

2. Dependency on Datasets and Prompts: The performance of the watermarking method seems to be influenced by the given prompts and datasets used in the experiments. For instance, watermarking the output to factual questions with limited flexibility in answers is noted as challenging.

3. Focus on Text Completion Tasks: The evaluation of the watermarking method is mainly conducted in the context of text completion tasks. The applicability and effectiveness of the watermark across different downstream tasks, such as question-answering and summarization, are suggested as areas for future evaluation.

All the above limitations have been mentioned in future work, which implies the authors still have substantial work to complete to make the results more convincing.

**Questions:**

1. Can the proposed method be extended to watermark multiple models?
2. The performance of the proposed method is not significantly better than the baseline method and is influenced by various factors such as the models used, the datasets, and the given prompts. I doubt the generalizability of the proposed method based on the current experiments.

---

> ### Author Response · Authors · 2023-11-23
>
> We thank the reviewer for the insightful reviews.
>
> “Can the proposed method be extended to watermark multiple models?”
> As also the reviewer expresses the model-agnostic nature of our approach as one of the strengths of our paper, our proposed approach is not a model specific watermarking method. It can be applied on any LLM.
>
> “The performance of the proposed method is not significantly better than the baseline method and is influenced by various factors such as the models used, the datasets, and the given prompts. I doubt the generalizability of the proposed method based on the current experiments.”
>
> We agree that more datasets and models will be useful for more conclusive findings. However,  due to space limitations and the cost of experiments (in terms of time and hardware requirements), we could only use three different models and two different datasets to increase the reliability of our results. We also conducted experiments in which we vary the temperature under generation. This allowed us to evaluate our methods with various probability distribution entropies. We would like to note that our experiments with various temperature values also address the need of using different tasks. For instance, in the task of answering factual questions, we expect that the entropy of the probability distribution will be low while we expect the opposite (i.e., high entropy) when we utilize LLMS to generate stories without any restriction to be factual.

---

### Official Review · Reviewer_b2Mo · 2023-11-02

**Soundness:** 2 fair
**Presentation:** 3 good
**Contribution:** 1 poor
**Rating:** 3
**Confidence:** 4

**Summary:**

This paper proposes a decoding procedure for embedding a statistical watermark in samples from an autoregressive language model.

The idea is to sample multiple candidate tokens from the model at each step i, and choose the candidate that maximizes a random number X_i generated by seeding a PRNG with the SHA256 hash of the candidate token together with the previous k tokens in the sequence. Text decoded in this way can be detected by calculating the average average of X_i, and testing the hypothesis that this mean deviates from the null hypothesis (no maximization). There are several hyper-parameters to this algorithm: the length of the hash sequence, the number of resamples, and whether to sample with or without replacement.

Different hyper-parameter configurations of the proposed method are compared to a baseline watermarking procedure proposed by Kirchenbauer et. al. using the OPT-1.3B, BTLM-3B, and Llama2-7B models.

**Strengths:**

The proposed method is easy to understand and simple to implement.

**Weaknesses:**

The setting (watermarking, robustness to attacks) and methodology (decoding algorithms, hashing) of this paper are quite similar to Kirchebauer et al., and I am not convinced that the newly proposed method is an significant improvement over the Kirchenbauer baseline. There is no formal analysis of the proposed watermark, and the experimental results are a step back compared to the breadth of evaluations and attacks presented in the Kirchenbauer baseline paper. Notably missing here are studies of watermark strength as a function of sequence length, and robustness beyond a simple substitution attack.

The discussion of the proposed watermark suffers from both overclaiming and insufficient analysis. "In our work, we interfere the sampling process without changing LLMs’ probability distribution over vocabulary while Kirchenbauer et al. (2023a) interfere the probability distribution." This is not true. The proposed algorithm is a uniform resampling over candidate tokens (assuming the SHA256 hash behaves well) which clearly changes the distribution; this change might be amenable to a clean mathematical description.

I am not fully convinced of the metrics chosen to evaluate the quality of generated text. Why is a paraphrasing similarity model (P-SP) being used to evaluate generation quality? Is the premise that the generated text to be similar/paraphrasing of the human text; isn't this contrary to the premise of open-ended text generation? This seems like a misapplication of P-SP. Why not use, e.g., sample perplexity under a larger LM (as used as a proxy for quality in the Kirchenbauer watermarking paper).

The detectability results in Table 1 emphasize a regime where all proposed watermarks work well. The claim of superior detectability vs. the Kirchenbauer watermark is based on z-scores of ~17 vs ~10. In the more challenging paraphrasing attack setting, there seems to be a significant performance advantage only for sampling without replacement (Figure 1; SWOR). But if we believe the proposed sample quality metrics, SWOR causes significant degradation in sample quality (Table 3). If anything, I suspect the metrics underestimate the degradation caused by SWOR: for low-entropy predictions, it forces the model to sample uniformly among unlikely candidates which seems quite bad.

**Questions:**

Why do the reports of experimental results distinguish between a SWR detector and a SWOR detector? Aren't these the same algorithm?

---

> ### Author Response · Authors · 2023-11-23
>
> We thank the reviewer for the insightful reviews.  We appreciate your feedback and have taken your comments into careful consideration. Below, we address some of your concerns and questions:
>
> Firstly, we realized that we were not clear enough to emphasize the differences between SWR and SWOR approaches. When in sampling without replacement (SWOR) approach, we ensure that we select y different candidate tokens. Therefore their corresponding secret numbers will be distinct. In contrast, sampling with replacement (SWR) approach allows a token to be selected multiple times, and there is no assurance of uniqueness among the y candidate tokens. This becomes relevant when dealing with distributions of low entropy, where the probability mass is heavily concentrated on one or a few tokens. In such cases, SWR may repeatedly sample the same token, reducing the number of distinct candidate tokens.
>
> “....Why is a paraphrasing similarity model (P-SP) being used to evaluate generation quality?”
> Regarding the usage of paraphrasing similarity model evaluation, we use the paraphrasing similarity model (P-SP) because it was also used in prior work on watermarking, e.g., Kirchenbauer et al. (2023b).

---

### Official Review · Reviewer_bSV5 · 2023-11-02

**Soundness:** 2 fair
**Presentation:** 2 fair
**Contribution:** 2 fair
**Rating:** 3
**Confidence:** 2

**Summary:**

The authors propose a watermarking scheme by interfering with the randomness of generating the next token. The sampling watermarkers first multinomially sample some tokens and then choose the token that can maximize the secret number as the next token. Experiments show that the watermark is detectable and robust against token-level paraphrasing attacks.

**Strengths:**

1. The proposed watermarking scheme offers a way to compute the statistical confidence interval to analyze the sensitivity of the watermark.
2. Experiment results reveal that the watermarking scheme only slightly decreases the quality of the generated text.
3. The watermarking scheme is also robust against token-level paraphrasing attacks.

**Weaknesses:**

1. The main idea of this approach seems very similar to the one proposed by Kirchenbauer et al. (2023a). The algorithm in [Kirchenbauer 2023a]:
i. Compute the probability distribution of the next token
ii. Use the previous tokens and a hash function to randomly partition the vocabulary into
“green list” and “red list”
iii. Modify the probability distribution and then sample the next token
The algorithm proposed in this paper:
i. Compute the probability distribution of the next token
ii. Use the previous tokens and a hash function to randomly generate the secret number for the candidate tokens.
iii. Choose the next token based on the secret number
If we treat the sampled candidate tokens as “green list” and other tokens as “red list”, then these two algorithms are very similar. The proposed approach just changes the partition method and the hash function. It would be better if the authors could provide more intuitions about what is the main difference between the proposed approach and that in [Kirchenbauer et al. (2023a)]. Otherwise, the contribution of this work seems limited.

2. The authors only evaluate the robustness of text by text substitution attack. This work would be better if the authors could evaluate the robustness against text deletion and text insertion attacks.

3. This approach does not have a factor that can control the strength of the watermark. It always chooses the candidate token that can maximize the secret number. If we can control the strength of the watermark injection, then we can balance the tradeoff between the quality of generated watermarked text and the strength of the watermark.

**Questions:**

1. What is the main difference between the proposed approach with Kirchenbauer et al. (2023a)?  The proposed approach does not interfere with the probability distribution of LLM, but it chooses the tokens according to their secret numbers instead of the original probability distribution. Is this just another way to change the probability distribution implicitly because the proposed approach adds some “logits” to the token with the largest secret number?
2. Why does the proposed approach work better than Kirchenbauer et al. (2023a)? Instead of changing the interference from a probability distribution to the sampling process, are there other reasons the proposed approach is better?

---

> ### Author Response · Authors · 2023-11-23
>
> We thank the reviewer for the insightful reviews. We understand that we should have put more emphasis on the distinctions between our approach and Kirchenbauer et al.’s approach. We first would like to note that all watermarking methods for LLMs would need to somehow affect probability distributions. Therefore, the main difference between watermarking methods for LLMs will be how these probability distributions are changed and the impact of these changes on the resultant texts. Now we would like to highlight these differences.
>
> 1) In Kirchenbauer  et. al.’s  approach, after the probability distribution for the next token is computed, a randomly selected subset of tokens (depending on the previous token(s)) are promoted.  In contrast, in our approach we sample from the distribution y (the sampling count) times. From these y candidate tokens we pick one that maximizes the secret number (essentially uniformly sampling from the candidate tokens). We would like to note that the candidate tokens are all tokens that could have been sampled by the sampler in the first place.
>
> 2) The reviewer mentioned that our candidate tokens can be considered as “green list” and other tokens as “red list”, similar to Kirchenbauer  et. al.’s  approach. We kindly disagree with  this conclusion because there is a slight difference: We pick the token that maximizes the secret number from the y candidate tokens which are selected based on only LLM’s probability distribution (i.e., without any promotion of randomly selected tokens). We don’t pick tokens that are not in this set. We try to guarantee the “quality” of the y candidate tokens by directly sampling from the distribution.
>
> 3) Another distinction between Kirchenbauer 2023a’s approach and ours is the way we handle low entropy distributions. If the probability mass is concentrated in a single or few tokens, in our approach (SWR), sampling from the distribution is done “with replacement” meaning the y candidate tokens could all be the same token. In such a case, there would be only a single distinct candidate token which is picked regardless of its secret number. This token is essentially “unwatermarked”. The effect of this is shown in Table 4, in which we artificially manipulate the entropy of the distributions using temperature. As shown, the watermark is less detectable when the underlying distributions have less entropy. In Kirchenbauer et. al.’s  approach, we randomly promote some tokens, after this we sample from this distribution. If tokens from the “red list” capture a significant probability mass, they can potentially be selected. In Table 4, the watermark is less detectable when the underlying distributions are of lower entropy. However, SWOR variant of our approach in which the detectability is independent from the underlying distributions entropy is not affected by the entropy of the probability distribution. This is because we ensure y candidate tokens to be distinct by sampling without replacement, thus we always select y different candidates.
> 4) We also realized that we were not clear about how to control  the strength of the watermark. The sampling count has a direct impact on the strength of the watermark. In particular, as shown in Table 3 and discussed in section 5.2.4 as we increase the sampling count y, there are more distinct candidate tokens to maximize the secret number from. Maximizing from more distinct values results in a higher overall secret number value which results in a more detectable watermark.

---

### Meta-Review · Area_Chair_Meaq · 2023-11-27

**Metareview:**

All the reviewers have concerns about the paper and are not in favor of acceptance. Most prominently, the comparison with Kirchenbauer et al. (2023a). The rebuttal period did not clear out the concerns.

**Justification For Why Not Higher Score:**

All the reviewers are negative about the paper.

**Justification For Why Not Lower Score:**

N/A

---

### Decision · Program_Chairs · 2024-01-16

Reject